# Amount Estimation Method for Food Intake Based on Color and Depth Images through Deep Learning

**DOI:** 10.3390/s24072044

**Published:** 2024-03-22

**Authors:** Dong-seok Lee, Soon-kak Kwon

**Affiliations:** 1AI Grand ICT Center, Dong-Eui University, Busan 47340, Republic of Korea; ulsan333@gmail.com; 2Department of Computer Software Engineering, Dong-Eui University, Busan 47340, Republic of Korea

**Keywords:** food intake amount estimation, volume estimation, RGB-D image, object detection, deep learning

## Abstract

In this paper, we propose an amount estimation method for food intake based on both color and depth images. Two pairs of color and depth images are captured pre- and post-meals. The pre- and post-meal color images are employed to detect food types and food existence regions using Mask R-CNN. The post-meal color image is spatially transformed to match the food region locations between the pre- and post-meal color images. The same transformation is also performed on the post-meal depth image. The pixel values of the post-meal depth image are compensated to reflect 3D position changes caused by the image transformation. In both the pre- and post-meal depth images, a space volume for each food region is calculated by dividing the space between the food surfaces and the camera into multiple tetrahedra. The food intake amounts are estimated as the difference in space volumes calculated from the pre- and post-meal depth images. From the simulation results, we verify that the proposed method estimates the food intake amount with an error of up to 2.2%.

## 1. Introduction

Food intake measurement is an important factor in diet management. However, it is inaccurate to manually measure food intake amounts due to subjective bias [1]. An automated method allows increasing the accuracy of food intake amount measurement. Food intake amount can be measured by sensing sounds, body movement, or changes in meal table weight during a meal [2,3,4,5] through mounting a device on a body or a table. However, the methods cannot classify food types. A color image enables detecting the food and estimating the food intake amount. The food objects are accurately detected from the color image through deep-learning networks such as Faster R-CNN [6], Mask R-CNN [7], and YOLO [8]. Several studies have utilized deep-learning networks to classify food types based on color image [9,10,11,12,13]. However, food volume measurement based on color images is challenging work because color images lack spatial information for reconstructing 3D scenes. The following volume estimation studies for food have been conducted to overcome the limitations of volume measurement based on color image utilizing a reference object with a known size and a volume [14,15,16,17]; employing predefined 3D models for foods [18,19]; and acquiring 3D information through camera parameters [20,21,22]. However, the methods are hard to apply practically because they require prior knowledge about the size and shape of a reference object or a food container. Our previous study [19] estimated food intake amount using predefined 3D shapes depending on the food types. The method accurately classified food types and estimated the approximate food volume through color images. However, the method is not suitable for the precise amount estimation of food intake due to the limitations of food volume measurement based on color images. Depth images have distances between a camera and subjects in the capturing direction as pixel values. Depth images can provide additional 3D spatial information that is hard to detect in color images. Depth images can be additionally utilized to improve object localization, which involves finding the bounding boxes of objects, in various fields such as agriculture [23,24,25,26]. Object volume can also be measured using the 3D spatial information obtained from a depth image. Depth-image-based methods have been proposed for food volume estimation [27,28,29,30,31,32,33]. However, these methods need shape information about food containers. Furthermore, the majority of these methods focus on estimating food volume. In food intake amount estimation, it should additionally be considered that the estimation accuracy can be affected by differences in food location between pre- and post-meal images.

In this paper, we propose an amount estimation method for food intake based on both color and depth images without prior knowledge about foods or food containers. Two pairs of color and depth images capture the foods on a meal plate at pre- and post-meal stages. The color images are segmented into food regions through Mask R-CNN. The post-meal color image is transformed to match the food region locations between the two color images. The same transformation is applied to the post-meal depth image. The pixel values of the post-meal depth image are compensated depending on the image transformation. For each depth image, the space volume is calculated between the camera and the food surface for each food region. The food intake amount is estimated as the space volume difference between the pre- and post-meal depth images.

The contributions of the proposed method are as follows: We propose an amount estimation method for food intake based on color and depth images without any prior knowledge; we present an image transformation method for aligning a meal plate between pre- and post-meal images; we introduce a pixel compensation for the depth image after image transformation. The proposed method can be applied to food intake amount estimation in various situations.

## 2. Related Works on Amount Estimation for Food Intake

The automated estimation of food intake amount is classified into sensor-based and image-based methods. Food intake amount can be measured by sensing changes in meal table weight via a mounted pressure sensor [2,3]. Food intake amount can also be measured by sensing eating sound [4] or body movement [5] during a meal. These methods involve the inconvenience of having to wear a wearable device or mounted device at the meal table.

Amount estimation for food intake from a color image is challenging work because it is difficult to find the relationship between the region of an object in the color image and its 3D volume. The object’s volume can be estimated from the color image through a reference object, predefined 3D models, camera parameters, or information about object depths.

A reference object whose shape or volume is known in advance can be used to estimate the volume of another object in the same color image. The volume is estimated through the area ratio of the target object to the reference object. A food container such as a dish or a bowl can be designated as a reference object for volume measurements. Okamoto and Yanai [14] estimate the food volume by multiplying the actual volume of the reference object by the area ratio of the food to the dish. However, this method has inaccuracies in food volume estimation as it ignores the length of the camera capturing direction, called depth, during volume estimation. Volume estimation methods are proposed through the relation that is pre-modeled between the area size ratio and the volume ratio. Hippocrate et al. [15] propose a volume estimation method through the relation between food volume and the width ratio of the food to the dish. Jia et al. [16] model the relation between the area size ratio of the food to the bowl and the food volume. Liu et al. [17] calculate the relation between the volume and the object regions through a trained convolutional neural network. However, volume estimation methods based on a reference object have a limitation in that the food needs to be placed in designated containers.

The estimation method based on predefined 3D models treats food shapes as simple models such as sphere, cylinder, and cuboid. The food volume is calculated through a volume equation corresponding to its 3D model. Smith et al. [18] calculate food volume by directly specifying 3D food shape. Our previous study [19] determines 3D food shapes through food types which are classified by a deep-learning object detector. These methods have low estimation accuracies for foods that have shapes different from the predefined models.

Objects in a color image can be reconstructed into 3D shapes through intrinsic and extrinsic camera parameters. The intrinsic parameters explain the projection of subjects to the color image. The extrinsic parameters indicate a camera position and a camera capturing direction. The camera parameters can be found by detecting an image projection distortion through a known object shape. Fang et al. [20] and Yue et al. [21] find the camera parameters referring to the known shape of a food container in order to estimate food volumes. Steinbrener et al. [22] estimate the camera parameters through CNN and LSTM layers using food images captured from various positions.

The difficulty of volume estimation based on a color image is due to lacking spatial information about the z-axis, that is, the camera capturing direction. A depth image stores z-axis distances from a camera as pixels. The 3D coordinates of the pixels are calculated through their values and 2D coordinates in a depth image. Object size and volume can be measured by calculating distances between key points such as corners through their 3D coordinates [34,35]. Lu et al. [27] generate a point cloud that consists of 3D points from a depth image for reconstructing the 3D surfaces of food and table. Food volumes are estimated by calculating space volumes between the foods and the table. Meyers et al. [28] convert the depth image into a voxel representation and then grids it into squares with a certain area. The volume of each grid is calculated through its area and the average of pixel values.

## 3. Amount Estimation for Food Intake Based on Color and Depth Images

Food intake amount can be estimated by comparing food volumes that are estimated from pre- and post-meal images. In image-based methods, larger food volumes are estimated due to including the volumes of spaces obscured by food containers. The previous methods reduce the estimation error through a prior knowledge of food container shapes [15,16,17] or by assuming its side to be vertical [27,28]. However, these methods have a limitation since the prior knowledge or the assumption sometimes differs from the actual food containers. From the perspective of food intake estimation, the obscured space is measured as the same volume in both the pre- and post-meal images if the food container’s locations are matched in the two images. Therefore, the space volume can be offset as shown in Figure 1, then only the food intake is measured. However, it is reasonable to assume that the food container’s locations often differ between the two images. In this case, the obscured space is not accurately offset because the volumes measured in the two images are different. In order to solve this problem, we present an image transformation method to match the container locations.

In this paper, we propose an amount estimation method for food intake based on both color and depth images captured by an RGB-D camera (RealSense D415 manufactured by Intel, Santa Clara, CA, USA). In this paper, we refer to object segmentation for food regions as ‘region detection’. A color image is utilized to detect food types and regions. For depth images, food intake amounts are estimated in the detected food regions. This paper deals with foods on a meal plate as shown in Figure 2. However, this does not mean that the proposed method can only apply to the foods placed in a certain container. The proposed method can estimate food intake amounts for any food container trained in a deep-learning network.

In order to estimate food intake amounts, foods in a meal plate are captured for both color and depth images at pre- and post-meals. In the color images, Mask R-CNN, which is a deep-learning network for object segmentation, detects a meal plate region and food regions in pixel units. The food regions in the pre-meal color image are designated as regions of interest (ROIs) for food intake amount estimation. Subsequently, the post-meal color image is transformed to align the food regions of the pre- and post-meal color images. The same transformation is applied to the post-meal depth image. The pixel values of the post-meal depth image are adjusted for pixel compensation after the image transformation. For the depth images, 3D points corresponding to the pixels in the ROIs are obtained through a pinhole camera model. A space volume between the food surface and the camera is calculated through the 3D points. The food intake amount is estimated by comparing the space volumes between the pre- and post-meal depth images. Figure 3 shows the flow of the proposed method.

### 3.1. Detections of Foods and Meal Plate through Deep-Learning Object Detector

To accurately estimate food intake amounts, the food regions need to be precisely detected. Deep-learning networks can effectively detect objects, including foods [19]. Mask R-CNN [7] is employed to detect foods at the pixel level. Mask R-CNN’s backbone, ResNet-50 [36], extracts features from the color images. Regions for detecting food are found through the region proposal network (RPN) and ROIAlign. The food types and regions are detected through a fully convolutional network (FCN) and fully connected layers (FC layers), respectively. Figure 4 illustrates the flow of food region detection through Mask R-CNN.

For training Mask R-CNN, a dataset is generated by capturing color images including the Korean foods on the meal plate by ourselves. The meal plate includes a main dish, which is steamed rice, five side dishes, and soup in a bowl. The dataset consists of the pre-meal color images and the post-meal color images with 25%, 50%, and 75% food intakes as shown in Figure 5. For a total of 522 color images in the dataset, 460 color images are utilized to train the object detector and the others are used to verify it. We apply image blurring, image rotation, and image flip for data augmentation. Epochs for network training are set to 60,000 times. The region detection results for the foods and meal plate are shown in Figure 6.

### 3.2. Image Transformation and Correction for Food Intake Amount Estimation

In order to align the meal plate regions of the two images, homography transformation based on the meal plate regions is applied to the post-meal color image. In homography transformation, new image coordinates $\left(\tilde{x},\;\tilde{y}\right)$ are calculated through a 3 × 3 matrix **H** as follows:(1)$$\begin{array}{c} AH=B\\ A=\left[\begin{array}{c} \tilde{x}\\ \tilde{y}\\ 1 \end{array}\right]\;H=\left[\begin{array}{ccc} h_{1} & h_{2} & h_{3}\\ h_{4} & h_{5} & h_{6}\\ h_{7} & h_{8} & 1 \end{array}\right]B=\left[\begin{array}{c} x\\ y\\ 1 \end{array}\right], \end{array}$$
where **A** and **B** are the matrices of image coordinates after and before transformation, respectively. Four pairs of corresponding points between two images are required for finding the eight elements in **H**. The following equation is a transformation of (1) obtained by substituting the point pairs $\left(x_{k},y_{k}\right)$ and $\left({\tilde{x}}_{k},{\tilde{y}}_{k}\right)$ (*k* = 1, 2, 3, 4):(2)$$\begin{array}{c} ah=b\\ a=\left[\begin{array}{cccccccc} x_{1} & y_{1} & 1 & 0 & 0 & 0 & -x_{1}{\tilde{x}}_{1} & -y_{1}{\tilde{y}}_{1}\\ 0 & 0 & 0 & x_{1} & y_{1} & 1 & -x_{1}{\tilde{x}}_{1} & -y_{1}{\tilde{y}}_{1}\\ x_{2} & y_{2} & 1 & 0 & 0 & 0 & -x_{2}{\tilde{x}}_{2} & -y_{2}{\tilde{y}}_{2}\\ 0 & 0 & 0 & x_{2} & y_{2} & 1 & -x_{2}{\tilde{x}}_{2} & -y_{2}{\tilde{y}}_{2}\\ x_{3} & y_{3} & 1 & 0 & 0 & 0 & -x_{3}{\tilde{x}}_{3} & -y_{3}{\tilde{y}}_{3}\\ 0 & 0 & 0 & x_{3} & y_{3} & 1 & -x_{3}{\tilde{x}}_{3} & -y_{3}{\tilde{y}}_{3}\\ x_{4} & y_{4} & 1 & 0 & 0 & 0 & -x_{4}{\tilde{x}}_{4} & -y_{4}{\tilde{y}}_{4}\\ 0 & 0 & 0 & x_{4} & y_{4} & 1 & -x_{1}{\tilde{x}}_{1} & -y_{4}{\tilde{y}}_{4} \end{array}\right] \end{array}\;h=\left[\begin{array}{c} h_{1}\\ h_{2}\\ h_{3}\\ h_{4}\\ h_{5}\\ h_{6}\\ h_{7}\\ h_{8} \end{array}\right]b=\left[\begin{array}{c} {\tilde{x}}_{1}\\ {\tilde{y}}_{1}\\ {\tilde{x}}_{2}\\ {\tilde{y}}_{2}\\ {\tilde{x}}_{3}\\ {\tilde{y}}_{3}\\ {\tilde{x}}_{4}\\ {\tilde{y}}_{4} \end{array}\right].$$
**h** is obtained as follows:(3)$$h=a^{-1}b.$$

In order to match the meal plate regions in the two color images through homography transformation, the four corner points of the meal plate are found by a rotating calipers algorithm [37] for each color image as shown in Figure 7. **h** in (2) is calculated through the four pairs of the corner points between the two color images. Figure 8 shows the image transformation results. The meal plate regions are matched in the two images.

The same transformation is also performed on the post-meal depth image. For the depth image, the image transformation may lead to pixels having weird values. In other words, the pixel values in food regions may be larger than those of the table because of the image transformation. It implies that the food is positioned under the table as illustrated in Figure 9. The distance between the table and the foods is preserved even if the meal plate’s location is changed. Therefore, the pixel values can be compensated based on the depth values of the table. In order to obtain the depth values of the table, the table without any objects is captured in a depth image in advance. The pixel value of the post-meal depth image is compensated after applying the image transformation as follows:(4)$$\tilde{d}\left({\tilde{x}}_{k},{\tilde{y}}_{k}\right)=d\left(x_{k},y_{k}\right)-\left(b\left(x_{k},y_{k}\right)-b\left({\tilde{x}}_{k},{\tilde{y}}_{k}\right)\right),$$
where $d\left(\cdot \right)$ and $\tilde{d}\left(\cdot \right)$ are the pixel values before and after the compensation, respectively, and $b\left(\cdot \right)$ is the pixel values of the depth image that captures only the table in advance.

A depth image has more errors compared to a color image. Figure 10 illustrates temporal errors for certain locations in 30 consecutive frames of depth images. The temporal errors of depth image are observed as changes in pixel values within a specific range around the actual pixel value. The correct pixel values are most frequently measured in the images. In order to correct the temporal errors, the pixel values are determined as the mode values among 30 consecutive frames. Pixels at object edges are occasionally not measured as shown in Figure 11a. In Figure 11a, black in red circles means unmeasured pixels. The unmeasured pixels are filled by applying a 7 × 7 mean filter as shown in Figure 11b.

### 3.3. Food Intake Amount Estimation Based on Depth Image

We propose the amount estimation method for food intake without any prior knowledge of the foods or container shapes. In the pre- and post-meal depth images, we measure a space volume between the food surface, which is formed by 3D points corresponding to the pixels in the ROI, and the camera point. The space volume is calculated by dividing it into several tetrahedra as shown in Figure 12.

For obtaining 3D points corresponding to the pixels in the ROI, their 3D coordinates are calculated through a pinhole camera model [38]. The pinhole camera model assumes an image as an imaginary plane in a 3D coordinate system where subjects are projected as shown in Figure 13. In the pinhole camera model, 3D coordinates for a pixel $\left(x,y\right)$ are obtained as follows:(5)$$\left[\begin{array}{c} X\\ Y\\ Z \end{array}\right]=d\left(x,y\right)\left[\begin{array}{c} \left(x-c_{x}\right)/f\\ \left(y-c_{y}\right)/f\\ 1 \end{array}\right],$$
where $\left(X,Y,Z\right)$ are 3D coordinates, $\left(c_{x},c_{y}\right)$ are optical center coordinates for the camera, and *f* is the camera focus length, which is the distance between the image plane and the camera. The optical center may be different from the image center.

For both the depth images of the pre- and post-meals, the pixels in the ROI are divided into multiple triangles as shown in Figure 14a for calculating the space volumes. The width and height of each triangle are one pixel unit, respectively. In each triangle, the 3D points corresponding to three pixels serve as vertices defining the base plane of a tetrahedron. All the tetrahedra share a common top vertex corresponding to the camera point. The *k*th tetrahedron volume is as follows:(6)$$V_{k}=\frac{1}{3}s_{k}h_{k},$$
where *s_k_* and *h_k_* are a base plane’s area and a distance to the origin, respectively. The *s_k_* is calculated from its vertices $\left(X_{i},Y_{i},Z_{i}\right)\;\left(i=1,\;2,\;3\right)$ as follows:(7)$$\begin{array}{c} s=\frac{1}{2}\left|v_{1}\times v_{2}\right|\;\;\\ v_{1}=\left(X_{2}-X_{1},\;Y_{2}-Y_{1},Z_{2}-Z_{1}\right)\;\\ v_{2}=\left(X_{3}-X_{1},\;Y_{3}-Y_{1},Z_{3}-Z_{1}\right), \end{array}\;$$
where × means a cross product operator. The value of *h_k_* is obtained as follows:(8)$$h_{k}=\frac{\left|p_{4}\right|}{\sqrt{p_{1}^{\;\;2}+p_{2}^{\;\;2}+p_{3}^{\;\;2}}}\;\begin{array}{l} p_{1}=Y_{1}\left(Z_{2}-Z_{3}\right)+Y_{2}\left(Z_{3}-Z_{1}\right)+Y_{3}\left(Z_{1}-Z_{2}\right);\\ p_{2}=Z_{1}\left(X_{2}-Z_{3}\right)+Z_{2}\left(X_{3}-X_{1}\right)+Z_{3}\left(X_{1}-X_{2}\right);\\ p_{3}=X_{1}\left(Y_{2}-Y_{3}\right)+X_{2}\left(Y_{3}-Y_{1}\right)+X_{3}\left(Y_{1}-Y_{2}\right);\\ p_{4}=X_{1}\left(Y_{2}Z_{3}-Y_{3}Z_{2}\right)+X_{2}\left(Y_{3}Z_{1}-Y_{1}Z_{3}\right)+X_{3}\left(Y_{1}Z_{2}-Y_{2}Z_{1}\right). \end{array}$$

The space volume between the camera and the food surface is calculated by summing the tetrahedron volumes. The food intake amount is estimated as the difference of the space volumes between the pre- and post-meal depth images.

## 4. Simulation Results

In our previous work [19], we evaluated Mask R-CNN [7] and YOLOv8 [39], which are deep-learning networks for region detection at the pixel level. The food classification accuracies are 97.8% and 95.6% for Mask R-CNN and YOLOv8, respectively. The detection accuracy for the food region was measured by calculating the intersection over union (IoU) as follows:(9)$$\mathrm{IoU}=\frac{n\left(G\cap D\right)}{n\left(G\cup D\right)},$$
where *G* and *D* are the ground truth and detected regions, respectively, and $n\left(\cdot \right)$ denotes the number of pixels in the region. The region detection accuracies were 94.1% and 86.4% for Mask R-CNN and YOLOv8, respectively.

For estimation evaluation of the proposed method about food intake amount, we captured various objects including foods as both color and depth images with resolutions of 640 × 480. The intrinsic parameters of the camera, denoted as *f*, *c_x_*, and *c_y_* in (5), are 598.05, 319.48, and 241.50, respectively. The camera, which is fixed at the top position, captures downward on the foods and the meal plate on a table. A distance between the camera and the table is 40 cm. We do not provide any prior shape knowledge of the food or the food container for the simulation.

In order to evaluate the volume estimation accuracies by the proposed method, we captured both color and depth images for simple-shaped objects as shown in Figure 15. Table 1 presents the actual sizes and volumes of the objects. Table 2 presents the volume estimation results for the objects. The volume estimation error tends to increase as the actual object volume becomes larger. The object shapes have no correlation with the volume estimation accuracies.

We evaluate the amount estimation accuracies for food intake as shown in Figure 16. The estimation accuracy of the proposed method is compared with our previous method [19], which estimates a food intake amount from two color images based on a prior knowledge of food shapes. Actual food volumes are measured by filling a food container, whose volume is already known, without leaving any empty space. Table 3 presents the amount estimation results for food intake. The proposed method has a maximum estimation error of about 2.2%. The amount estimation for food intake is less accurate than the volume estimation for simple-shaped objects due to the formation of spaces between the food pieces when the food is arranged on a dish. Additionally, the proposed method is less accurate for small-sized foods like ‘fried anchovies’. For the small-sized food, the estimation precision for the food intake is further affected by depth image noise. The proposed method has higher estimation accuracies for all the foods than the previous method. This implies that the anticipated food shape based on the prior knowledge frequently does not correspond to its actual shape. On the other hand, the proposed method does not suffer from the problem caused by inaccurate shape models for the food or container.

We evaluate the proposed method for foods in the meal plate as shown in Figure 17. The foods are captured in three pairs of color and depth images: one is at the pre-meal; the others are at post-meals with about 30% and 60% intakes, respectively. Table 4 presents amount estimation results for the food intake. The proposed method tends to estimate the small-sized foods as larger intake amounts. Additionally, the proposed method may have some estimation error depending on how the foods are stacked.

## 5. Conclusions

In this paper, we proposed the amount estimation method for food intake based on both color and depth images. The color and depth images were captured by an RGB-D camera for foods at both pre- and post-meals including the meal plate. In both the pre- and post-meal color images, Mask R-CNN detected the regions of the foods and the meal plate. ROIs for the food amount estimation were designated as the food regions in the pre-meal color image. In order to accurately estimate the food intake amount, the meal plate regions were aligned between the two color images by applying the image transform to the post-meal color image. The same transformation was applied to the post-meal depth image. For the post-meal depth image, the pixel values were adjusted in order to compensate after the transformation. For the pre- and post-meal depth images, the image coordinates of the pixels in the ROIs were converted to 3D coordinates to calculate the space volume between the food surface and the camera. The space volume was calculated by dividing it into several tetrahedra. The food intake amount was estimated as the difference of the space volume between the two depth images. The proposed method achieved the amount estimation error for food intake as up to 2.2%. The method does not need additional information such as the shapes of food containers or camera parameters for food intake estimation. Therefore, the proposed method can apply to wide fields of dietary analysis.

## Figures and Tables

**Figure 1 sensors-24-02044-f001:**
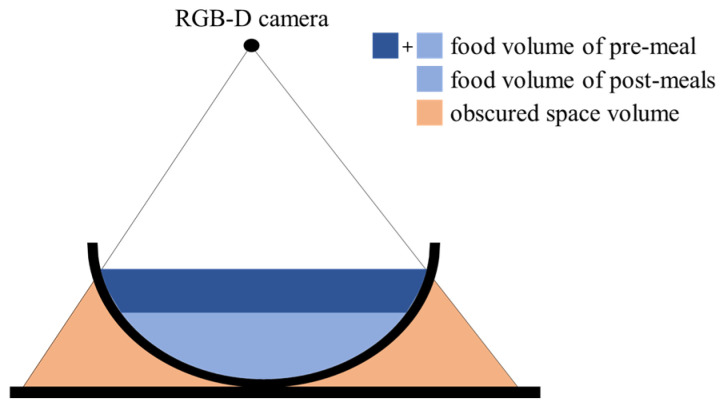
Removal of obscured space in amount estimation for food intake.

**Figure 2 sensors-24-02044-f002:**
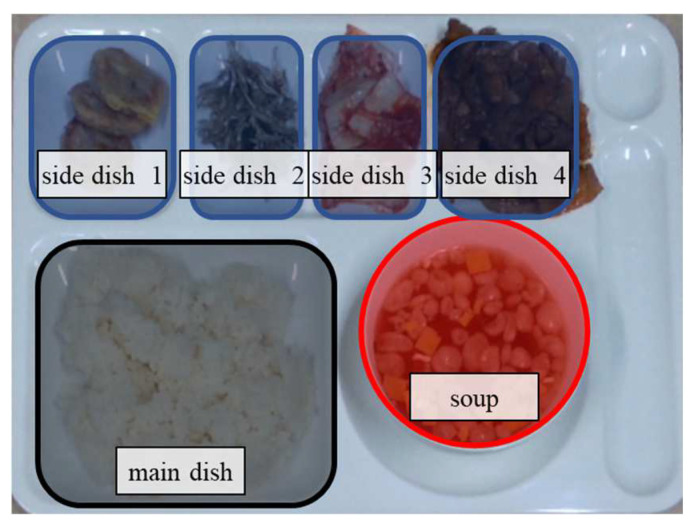
Foods and meal plate.

**Figure 3 sensors-24-02044-f003:**
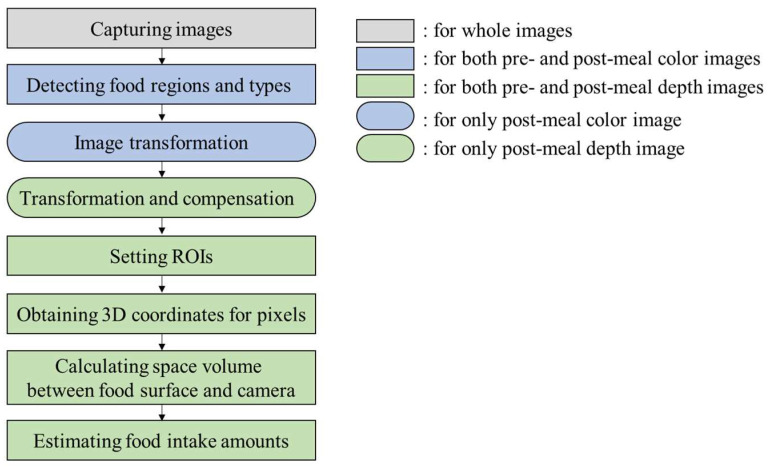
Flow of the proposed method.

**Figure 4 sensors-24-02044-f004:**
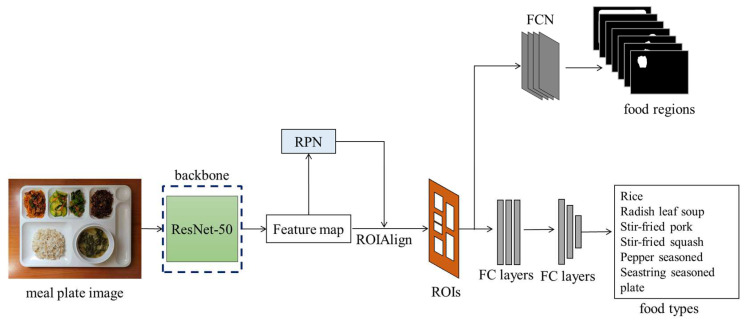
Flow of food detection and classification through Mask R-CNN.

**Figure 5 sensors-24-02044-f005:**
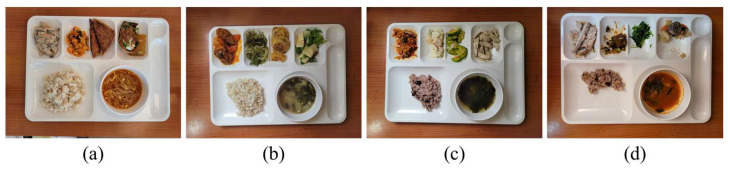
Dataset samples for detector training: (**a**) before meal; (**b**) 25% intake; (**c**) 50% intake; (**d**) 75% intake.

**Figure 6 sensors-24-02044-f006:**
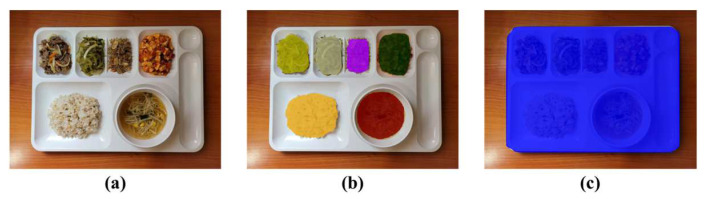
Region detection results through Mask R-CNN: (**a**) input image; (**b**) food regions; (**c**) meal plate region.

**Figure 7 sensors-24-02044-f007:**
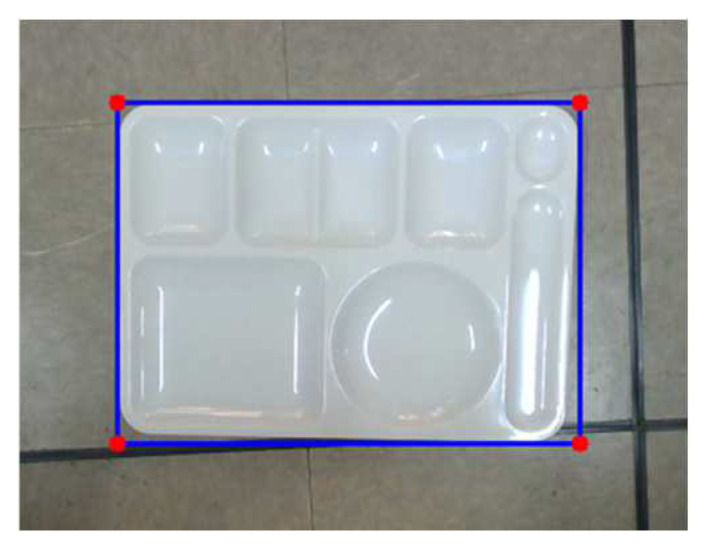
Finding four corner points of the meal plate for image transformation.

**Figure 8 sensors-24-02044-f008:**
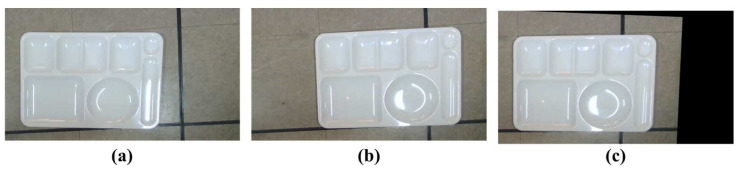
Image transformation results: (**a**) pre-meal image; (**b**) post-meal color image before transformation; (**c**) post-meal color image after transformation.

**Figure 9 sensors-24-02044-f009:**
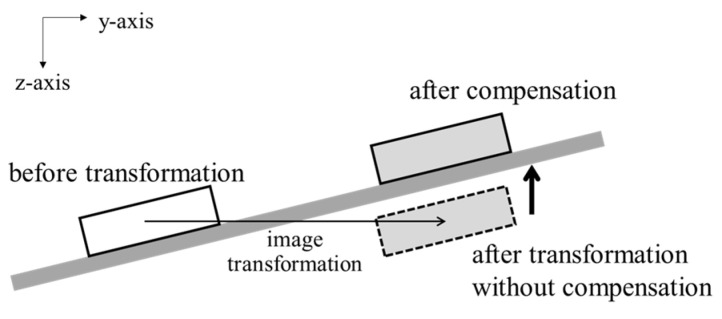
Compensation for pixel values of depth image after image transformation.

**Figure 10 sensors-24-02044-f010:**
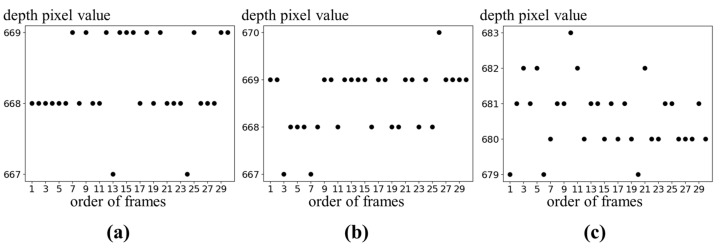
Temporal errors in depth image: (**a**) center; (**b**) left-top; (**c**) object edge.

**Figure 11 sensors-24-02044-f011:**
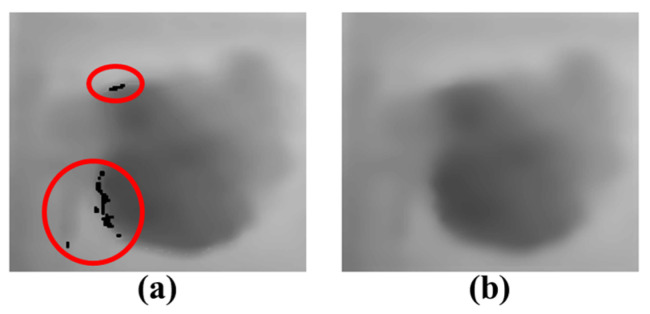
Depth image correction for unmeasured pixels: (**a**) unmeasured pixels in the depth image; (**b**) correction result through mean filter.

**Figure 12 sensors-24-02044-f012:**
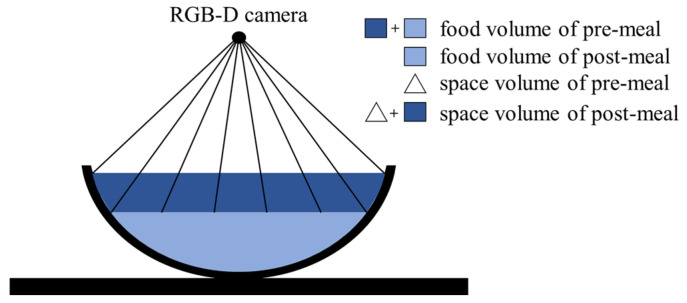
The proposed amount estimation for food intake through tetrahedra.

**Figure 13 sensors-24-02044-f013:**
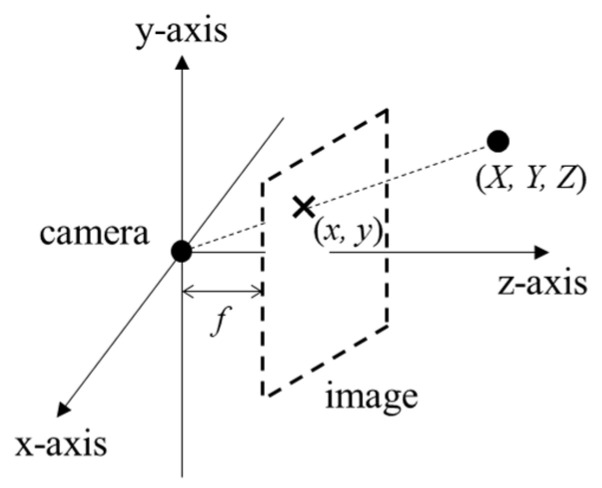
Projection onto image in a pinhole camera model.

**Figure 14 sensors-24-02044-f014:**
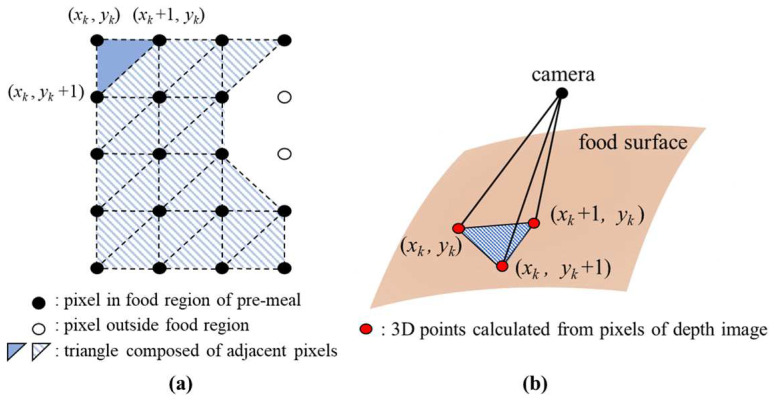
Calculating the space volume between the camera point and food surface through divided tetrahedra: (**a**) dividing pixels in the food region into multiple triangles; (**b**) tetrahedron formed by three neighboring pixels.

**Figure 15 sensors-24-02044-f015:**
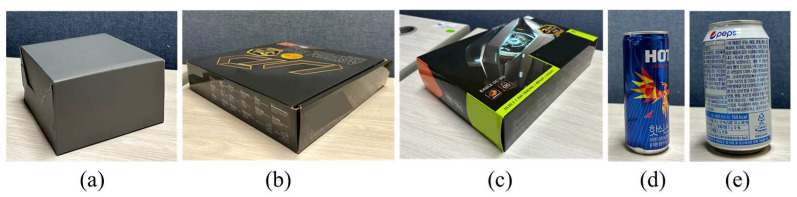
Objects for volume estimation simulations: (**a**–**c**) objects with a cuboid shape; (**d**,**e**) objects with a cylinder shape.

**Figure 16 sensors-24-02044-f016:**
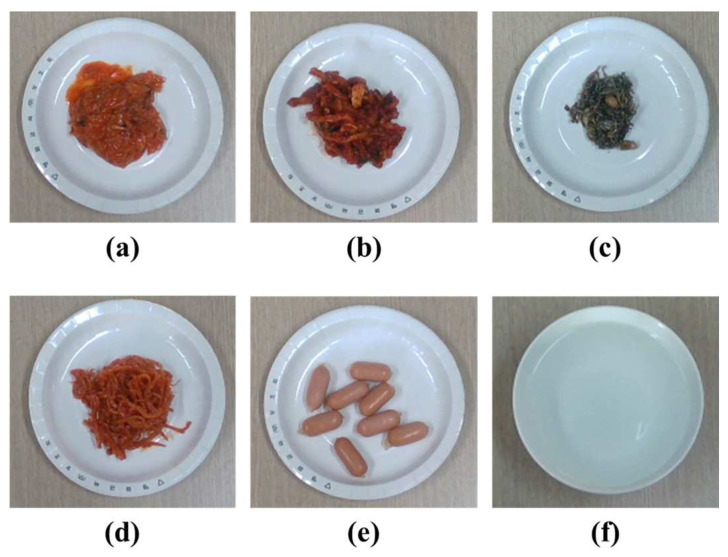
Foods for volume estimation simulation: (**a**) kimchi; (**b**) dried radish; (**c**) fried anchovies; (**d**) spicy squid; (**e**) sausages; (**f**) water in bowl.

**Figure 17 sensors-24-02044-f017:**
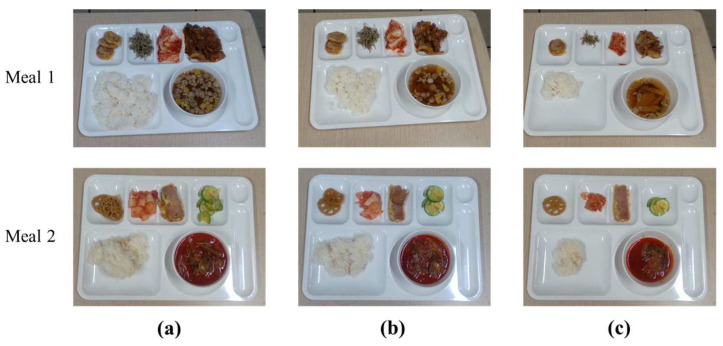
Object locations for volume estimation simulation: (**a**) pre-meal; (**b**) post-meal with 30% intake; (**c**) post-meal with 60% intake.

**Table 1 sensors-24-02044-t001:** Object information about shapes, sizes, and volumes.

Target	Shape	Size	Volume
(a)	cuboid	l: 12.4 cm, w: 12.6 cm, h: 7.5 cm	1171.80 cm^3^
(b)	cuboid	l: 25.0 cm, w: 25.0 cm, h: 5.0 cm	3125.00 cm^3^
(c)	cuboid	l: 20.0 cm, w: 30.0 cm, h: 7.0 cm	4200.00 cm^3^
(d)	cylinder	r: 4.6 cm, h: 13.2 cm	877.48 cm^3^
(e)	cylinder	r: 6.6 cm, h: 16.8 cm	2299.04 cm^3^

**Table 2 sensors-24-02044-t002:** Volume estimation accuracies for simple-shaped objects.

Target	Estimation Volume	Error Rate
(a)	1166.53 cm^3^	0.45%
(b)	3087.40 cm^3^	1.20%
(c)	4161.97 cm^3^	0.91%
(d)	873.89 cm^3^	0.41%
(e)	2282.83 cm^3^	0.71%

**Table 3 sensors-24-02044-t003:** Amount estimation accuracies for food intake.

Food Name	Actual Amount cm^3^			Estimated Intake Amount cm^3^ (Error Rate %)	
	Pre-Meal	Post-Meal	Food Intake	[19]	Proposed Method
kimchi	90.0	75.0	15.0	14.1 (6.0)	15.2 (1.3)
dried radish	105.0	90.0	15.0	13.7 (8.7)	14.8 (1.3)
fried anchovies	30.0	15.0	15.0	14.6 (2.7)	15.3 (2.2)
spicy squid	90.0	60.0	30.0	28.5 (5.0)	29.7 (1.0)
sausage	42.4	21.2	21.2	19.1 (9.9)	20.8 (1.9)
water	200.0	100.0	100.0	75.3 (24.7)	99.1 (0.9)
water	300.0	100.0	200.0	123.1 (38.5)	198.3 (0.8)

**Table 4 sensors-24-02044-t004:** Intake amounts estimated for foods in meal plate by the proposed method.

Target	Food Name	Estimation Amount		
		Pre-Meal	30% Intake	60% Intake
Meal 1	rice	390 cm^3^	169 cm^3^	308 cm^3^
	fish cake soup	270 cm^3^	67 cm^3^	146 cm^3^
	meatball	50 cm^3^	19 cm^3^	36 cm^3^
	fried anchovies	40 cm^3^	16 cm^3^	29 cm^3^
	kimchi	45 cm^3^	21 cm^3^	34 cm^3^
	stir-fried pork	175 cm^3^	46 cm^3^	112 cm^3^
Meal 2	rice	308 cm^3^	127 cm^3^	217 cm^3^
	spicy beef soup	308 cm^3^	113 cm^3^	218 cm^3^
	braised lotus roots	50 cm^3^	25 cm^3^	35 cm^3^
	kimchi	50 cm^3^	20 cm^3^	37 cm^3^
	egg-dipped sausage	50 cm^3^	14 cm^3^	24 cm^3^
	fried squash	40 cm^3^	11 cm^3^	23 cm^3^

## Data Availability

Data are contained within the article.

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
