# Peer review of "Amount Estimation Method for Food Intake Based on Color and Depth Images through Deep Learning"

_sensors, 2024, doi:10.3390/s24072044_

Round 1
Reviewer 1 Report
Comments and Suggestions for Authors
The paper proposes an amount estimation method for food intake based on paired color and depth images pre- and post- meal. The color images are for detection using Mask R-CNN. The depth images are for amount estimation. The space volume for each food region is calculated by dividing a space between the food surfaces and the camera into multiple tetrahedra. The food intake amounts are estimated as the difference in volumes calculated from the pre- and transformed post-meal depth images. Experiments show the proposed method has low error rate.
Strengths:
- The paper is well-written and nicely structured.
- Correcting the pixel values of the transformed depth images is somewhat novel.
Weaknesses:
- The biggest drawback is only presenting the experiment results of the proposed method and lacking comparison with other methods. One of the baselines should be calculating volume using Fig 11. [15-17] and volume calculation from depth images in [a](reference below) can also be baselines.
- The contribution of this paper is not clear and the novelty is limited. [a] has already estimated volumes from depth images in 2015.
- It seems both Fig 11 and 12 need the container shape / depth / edge information to accurately calculate the volume. Since the empty plate image is in the dataset, why is Fig 12 better than Fig 11? Experiments comparing them are needed.
Detailed questions and suggestions
- It would be better to analyze the quality of the transformation as there are 8 variables and only 4 pairs of points are used, which could be affected by noise.
- It would be better to analyze the relationship between error and the granularity of the tetrahedra when calculating volumes.
- When performing the simulation experiment, is prior knowledge (volume formula of cuboids and cylinders) used? As the object backside cannot be captured, the space between the object and the background could also be included in the volume.
- In table 3, why is the volume of water pre meal < post meal?
Reference:
[a] Meyers, Austin, et al. "Im2Calories: towards an automated mobile vision food diary." Proceedings of the IEEE international conference on computer vision. 2015.
Comments on the Quality of English LanguageThere are minor grammar errors (e.g. L126 “The color images utilized to detect the food types and regions.” -> are utilized) but they do not affect the overall reading.
Reviewer 2 Report
Comments and Suggestions for Authors
The authors propose an interesting algorithm, possessing novelty, to solve the problem of estimating the amount of food consumed. The paper utilizes modern computer vision techniques as well as depth map processing. Nevertheless, there are several shortcomings that need to be improved before the paper can be published:
1) The introduction could be strengthened, for example, by adding citations of works on computer vision in agriculture (10.1016/j.compag.2023.108036, 10.1016/j.heliyon.2023.e14722), including depth cameras (10.1016/j.compag.2020.105687, 10.3390/agronomy11091780)
2) Conceptually, Mask R-CNN solves the segmentation problem rather than the detection problem. The concept of detection is used in the text. An introduction should be given that "generally speaking the model results in pixel-by-pixel food selection, but we will use the term detection".
3) Table 3: in the calculation of water in the bowl, the volume after the meal increased. Please explain this result in more detail.
